# Comparative efficacy and acceptability of different antihypertensive drug classes for cardiovascular disease prevention: protocol for a systematic review and network meta-analysis

Heidi Jussil,[1] Anna Chaimani,[2] Bo Carlberg,[1] Mattias Brunström  [1]

¹Department of Public Health and Clinical Medicine, Umeå University, Umea, Sweden
²Research Center of Epidemiology and Statistics (CRESS-UMR1153), INSERM, INRA, Universite de Paris, Paris, Île-de-France, France

**Correspondence to**
Dr Mattias Brunström;
mattias.brunstrom@umu.se

## ABSTRACT

**Introduction** Clinical practice guidelines differ in their recommendations on first-line antihypertensive drug classes. No adequately powered randomised controlled trial have assessed all major drug classes against each other, and previous meta-analyses have mainly relied on pairwise meta-analyses for treatment comparisons.

**Methods and analysis** A systematic review and network meta-analysis will be carried out to assess the efficacy and acceptability of all major antihypertensive drug classes. PubMed and CENTRAL were searched on 21 February 2020 to identify randomised controlled trials with at least 1000 person-years of follow-up, assessing any antihypertensive agent against other agents or placebo. All trials fulfilling the inclusion criteria will be assessed for risk of bias using the second version of Cochrane's risk of bias assessment tool. The study selection process, risk of bias assessment and data extraction are done by two authors in duplicate. Relative risks from individual trials will be combined in pairwise meta-analyses; in the absence of important intransitivity, random-effects network meta-analysis will be performed. The primary outcome for efficacy will be major adverse cardiovascular events, whereas the primary acceptability outcome will be treatment discontinuation for any reason. Additional outcomes include all-cause mortality, cardiovascular mortality, stroke, myocardial infarction, heart failure and acute renal failure. The impact of differences within drug classes will be explored through alternative networks, including analysing thiazide-like and thiazide-type diuretics separately.

**Ethics and dissemination** This review will only process aggregated study level data and does not require ethical approval. The findings will be published in a peer-reviewed medical journal.

**PROSPERO registration number** CRD42020205482.

## Strengths and limitations of this study

► This systematic review and network meta-analysis will be the first contemporary review to assess the efficacy and acceptability of all major antihypertensive drug classes, incorporating direct and indirect evidence.

► Broad inclusion criteria in terms of patient characteristics ensure wide applicability of the study findings; potential differences between patient groups will be explored through subgroup and sensitivity analyses.

► Alternative networks, with different node definitions, will be used to explore differences within drug classes, specifically differences between thiazide-like and thiazide-type diuretics.

► The main limitation of this review is the use of study-level data, with no possibility to account for individual patient characteristics in the analyses.

(CVD) worldwide,[2] and that lowering blood pressure in the hypertensive range reduces the risk of CVD.[2 3]

Five major drug classes have been shown to reduce the risk of CVD compared with placebo or no treatment; ACE inhibitors (ACEi), angiotensin II receptor blockers (ARB), calcium channel blockers (CCB), diuretics and beta blockers (BB).[4 5] Although BBs have generally been abandoned as first-line therapy due to less effective stroke prevention compared with other agents,[6] current hypertension guidelines differ in the recommendations on what drug class to use as first-line therapy.[7–9] Whereas US guidelines recommend single therapy with ACEi, ARB, CCB or diuretics initially,[7] European guidelines recommended to start treatment with a renin–angiotensin–aldosterone system (RAAS) inhibitor (ACEi or ARB), in combination with a CCB or a diuretic.[8] The most

## INTRODUCTION

Hypertension (blood pressure ≥140/90 mm Hg) is estimated to affect more than one billion people globally.[1 2] Studies have shown that hypertension is the most important risk factor for death and cardiovascular disease

recent UK recommendations, from the National Institute for Health and Care Excellence, support RAAS inhibitors in young people and people with diabetes, whereas CCBs are considered the drug class of choice for elderly and people of African heritage.[9]

Even though many antihypertensive agents have been compared with one another or placebo separately, no adequately powered trial has compared all major drug classes head-to-head, assessing CVD outcomes. Several systematic reviews and pairwise meta-analyses aiming to assess this have been published, but all have limited their analyses to comparative trials assessing two or more agents against each other, with the exclusion of placebo-controlled trials.[5 10–12] Excluding placebo-controlled trials means exclusion of potentially valuable information gained through indirect comparisons, likely affecting the overall results.[13] Further, the two most cited meta-analyses supporting the view that all major drug classes are comparable have used pairwise meta-analyses, comparing each drug class to all other drug classes.[5 10] This implicitly assumes that all drug classes used for comparison are similar, which may conceal clinically important differences. The only contemporary network meta-analysis assessing different antihypertensive agents in the general population to date, is hard to interpret due to limitations in study selection, data extraction and risk of bias assessment.[14]

All major systematic reviews have used the conventional classification of antihypertensive agents described above.[4 5 10–12] However, categorisation of drug classes might play a critical role for the results of network meta-analyses. For example, the class of diuretics may be split into thiazide-type and thiazide-like diuretics. US and European guidelines equate these subclasses,[7 8] whereas the UK guidelines preferentially recommend thiazide-like diuretics.[9] The latter recommendation is supported by pharmacological differences,[15] as well as recent meta-analyses assessing differences in blood pressure lowering and CVD reduction.[16 17] Another categorisation of interest is that of RAAS inhibitors (ACEi and ARB). Although pharmacologically distinct, these classes act on the same physiological system, and evidence from clinical trials suggests no difference in efficacy between classes.[18] Third, CCBs can be subdivided into dihydropyridine and non-dihydropyridine agents, the former being highly selective for the vasculature whereas the latter also has antiarrhythmic properties.[19]

In summary, clinical practice guidelines differ in their recommendations on first-line antihypertensive treatment. This reflects the absence of head-to-head comparisons between major drug classes, and hence uncertainty regarding the comparative efficacy between classes. The aim and purpose of this study is to compare the efficacy and acceptability of different antihypertensive drug classes for CVD prevention through a systematic review and network meta-analysis, incorporating evidence from comparative as well as placebo-controlled trials. Further, we will explore the potential differences among different subclasses of diuretics, RAAS inhibitors and CCB.

## METHODS

We will perform a systematic review guided by the recommendations from the Cochrane Collaboration.[20] The protocol is written with guidance from the Preferred Reporting Items for Systematic Review and Meta-Analysis Protocols statement, with additional considerations specific for network meta-analyses,[21 22] and was registered in the PROSPERO database on 21 August 2020. Any important updates to the protocol will be amended to PROSPERO and/or described in the final report. The final report will follow the recommendations from the Preferred Reporting Items for Systematic Review and Meta-Analysis - Network Meta-Analysis (PRISMA-NMA) statement.[23]

### Eligibility criteria
#### Types of studies
Randomised controlled trials comparing any single antihypertensive agent, or combination of antihypertensive agents, against another agent, combination of agents, placebo or no treatment will be included. Trials using a prospective randomised open-label blinded endpoint (PROBE) design will be accepted, whereas cluster-randomised trials, crossover trials and trials of combined or complex interventions will be excluded. Trials assessing different treatment strategies, such as more intensive versus less intensive treatment, will also be excluded, because they convey no information on drug-specific or drug class-specific effects. We will restrict our analyses to trials with at least 1000 person-years of follow-up to avoid small trials not primarily assessing the effect of blood pressure lowering agents on cardiovascular events, but rather the effect of different agents on blood pressure levels. Hundreds of such trials exist; cardiovascular events are uncommon, and when present recorded as adverse events and therefore not properly adjudicated.

#### Types of participants
Trials will be included if participants are either primary preventive, have stable coronary artery disease, previous cerebrovascular disease, diabetes mellitus or mild chronic kidney disease (estimated glomerular filtration rate (eGFR) 60–89 mL/min/1.73 m$^2$). Because the main drug classes listed below (see the Types of intervention section) may be combined (except for ACEi+ARB; combination excluded), and prescribed in any order without important interactions on cardiovascular outcome level, we will include trials of treatment naïve patients and previously treated patients in the network meta-analysis. Studies including more than 50% of participants with heart failure, left ventricular dysfunction, acute myocardial infarction, macroalbuminuria, end-stage renal disease or chronic kidney disease stage 3 or higher (eGFR <60 mL/min/1.73 m$^2$) at baseline will be

excluded. The reason for these exclusions is that one or several of the included drug classes have specific effects in these patient groups, that possibly goes beyond blood pressure lowering.[7 8 24 25] Thus, the transitivity assumption (see the Evaluation of the transitivity assumption section) would be violated if such trials were combined with trials assessing similar agents in patient populations where such effects are absent. Although some would argue for specific effects of RAAS inhibitors in people with stable coronary artery disease, as well as in people with diabetes and mild chronic kidney disease, recent systematic reviews have not confirmed such an effect on cardiovascular outcomes.[18 26–28] Based on these findings, trials in people with stable coronary artery disease, diabetes and mild chronic kidney disease will be included in our main analysis, and excluded in sensitivity analyses to assess the impact on the overall results.

### Types of intervention

The main interventions of interest are antihypertensive drug classes which have previously been shown to reduce the risk of cardiovascular events; ACE, ARB, BB, CCB and diuretics. These are the main treatment options in clinical practice for the types of participants listed above (see the Types of participants section), and will therefore constitute the decision set of interventions in this review. Other drug classes, such as potassium-sparing diuretics, alpha-blockers, peripheral vasodilators and central-acting agents, as well as drug combinations including any of the above-mentioned drug classes, placebo and no treatment, will be included in the network as supplementary interventions to maximise the amount of evidence through indirect comparisons (figure 1).

### Outcome measures

The primary outcome for efficacy will be the composite outcome, major adverse cardiovascular events (MACE). Definitions of MACE vary to some extent across trials, but commonly include cardiovascular mortality, stroke and myocardial infarction. Inclusion of heart failure, revascularisation and peripheral artery disease differ across trials. If available, we prefer the strict definition using cardiovascular mortality, myocardial infarction and stroke. If other definitions of MACE have been applied, they will generally be accepted but appropriateness for inclusion will be assessed on a case-by-case basis. All deviations from the

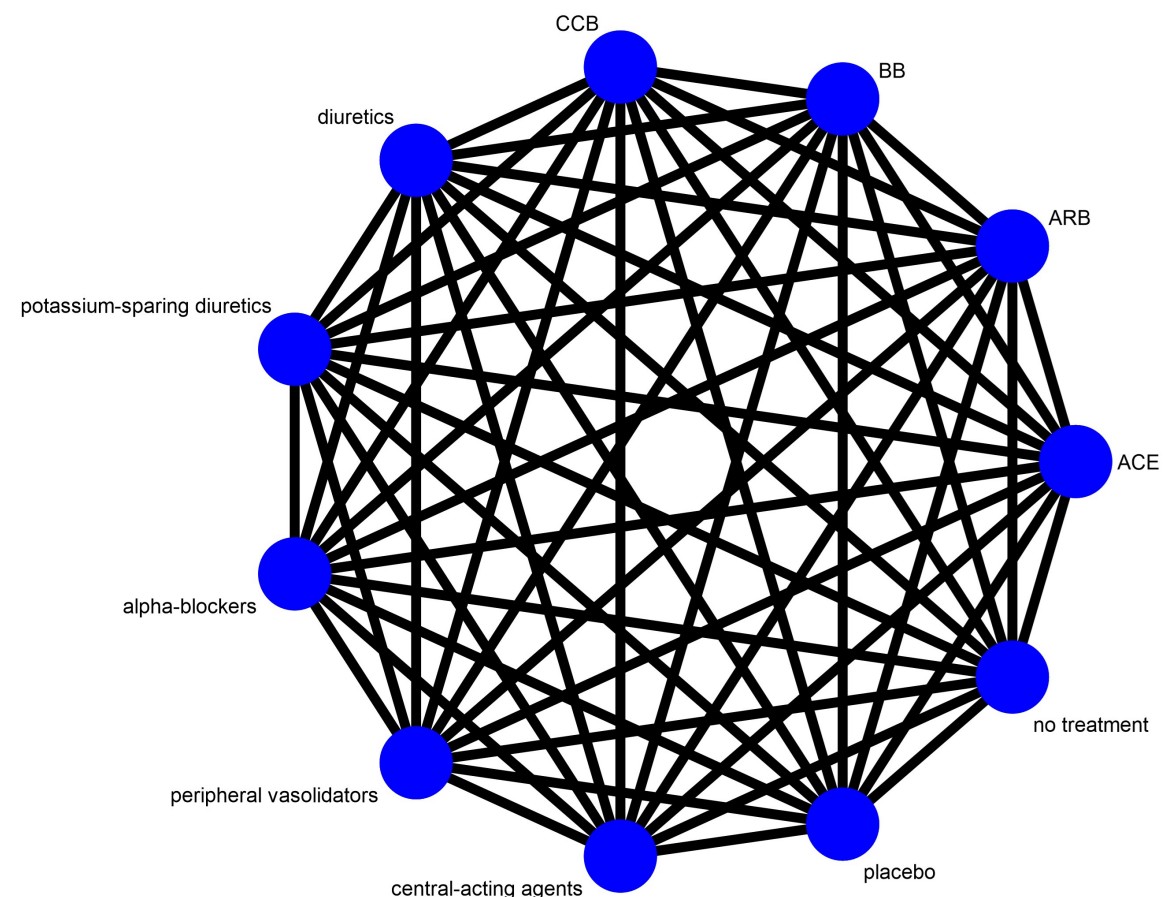

**Figure 1** Network of all possible comparisons according to conventional classification. ACEi, ACE inhibitors; ARB, angiotensin II receptor blockers; BB, beta blockers; CCB, calcium channel blockers.

preferred definition will be denoted, and the impact of MACE definition will be explored in sensitivity analyses, excluding trials not presenting results according to the preferred definition.

The primary outcome for acceptability will be permanent discontinuation of study medication. We will include discontinuations for any reason because this reflects safety as well as efficacy, and because the reporting of reasons for discontinuation differs to a large extent between trials.

Secondary outcomes are all-cause mortality, cardiovascular mortality, myocardial infarction, stroke, heart failure and acute kidney injury. Mortality outcomes, as well as myocardial infarction and stroke, have generally accepted universal definitions, although for myocardial infarction the definition has changed slightly over the years. For these outcomes, the definitions applied in individual trials will generally be accepted. For myocardial infarction and stroke we primarily aim to include fatal and non-fatal events, combined. If only fatal or non-fatal events are reported separately, however, we will accept and include the reported numbers. We will also accept 'acute coronary syndrome' as equivalent to myocardial infarction, but we will not include composite coronary events including revascularisations. For heart failure, we will accept definitions including new diagnosis of heart failure, heart failure hospitalisations and heart failure mortality. For acute kidney injury, we will apply the Kidney Disease: Improving Global Outcomes (KDIGO) definition, including at least 50% increase in serum creatinine, where appropriate. However, we will also include discontinuation of treatment due to kidney injury, as many trials were designed before the KDIGO definition of acute kidney injury was published.

### Search strategy

Preliminary searches in PubMed and CENTRAL were performed on 21 February 2020. We used search terms for blood pressure lowering treatment combined with several CVD terms in both databases. We restricted our search to randomised controlled trials, but applied no date or language restrictions.

Full search strategy in PubMed was (("antihypertensive agents"[Pharmacological Action] OR "antihypertensive agents"[MeSH Terms] OR "antihypertensive"[All Fields] OR "blood pressure-lowering"[All Fields] OR "blood pressure lowering"[All Fields] OR "blood-pressure lowering"[All Fields]) AND ("cardiovascular"[All Fields] OR "myocardial"[All Fields] OR "stroke"[MeSH Terms] OR "stroke"[All Fields] OR "heart failure"[All Fields] OR "mortality"[All Fields]) AND (randomized controlled trial[Publication Type] OR (randomized[Title/Abstract] AND controlled[Title/Abstract] AND trial[Title/Abstract]))).

Full search strategy for CENTRAL was (antihypertensive near agents OR "blood pressure-lowering" OR "blood pressure lowering" OR "blood-pressure lowering") AND ("cardiovascular" OR "myocardial" OR "stroke" OR "heart failure" OR "mortality") in All Text. Major

| Table 1 Variables for extraction | |
|---|---|
| **Descriptive (study level)** | **Analytical (treatment-arm level)** |
| Study ID | Study ID |
| Publication year | Treatment (category) |
| No. of participants randomised | Participants randomised (n) |
| Age (mean) | Participants followed for mortality (n) |
| Sex (% female) | Participants followed for CVD (n) |
| Baseline comorbidities | Follow-up duration (mean) |
| Cerebrovascular disease (%) | MACE (n) |
| Coronary artery disease (%) | Discontinuation (n) |
| Total CVD (%) | All-cause mortality (n) |
| Diabetes mellitus (%) | Cardiovascular mortality (n) |
| Intervention (drug and dose) | Stroke (n) |
| Control (drug and dose) | Myocardial infarction (n) |
| Funding | Heart failure (n) |
| Early termination | Acute kidney injury (n) |
| Reason for termination | Baseline SBP/DBP (mean) |
| Definition of MACE | Follow-up SBP/DBP (mean) |
| Definition other outcomes | SBP/DBP difference during follow-up (mean) |
| Comorbidity | |

.CVD, cardiovascular disease; DBP, diastolic blood pressure; ; MACE, major adverse cardiovascular events; SBP, systolic blood pressure.

systematic reviews and meta-analyses were further scrutinised for individual trials possibly missed by the database searches.[3 5 10 11 14 29–33]

Records from PubMed and CENTRAL were combined using EndNote reference software, removing duplicate records. Titles were screened by one reviewer (HJ) to remove apparently irrelevant publications. Abstracts and full-text articles were assessed against eligibility criteria by two reviewers independently (HJ and BC). Discrepancies on study inclusion were resolved by discussion or third-party involvement (MB). All studies fulfilling the above eligibility criteria will be reported in the systematic review, regardless of outcome reporting.

### Data extraction

Data will be extracted from original publications into Excel spreadsheets by two authors independently. Extracted data will include descriptive study characteristics as well as data for analytical purposes. A preliminary list of variables to extract is presented in table 1.

Differences in extracted data will be assessed going back to original publications and, if uncertainty

prevails, resolved by discussion or involvement of a third investigator.

## Blood pressure data

Mean systolic and diastolic blood pressure level at baseline and during follow-up will be collected for each treatment arm in each trial. If baseline blood pressure levels are not presented for each treatment arm separately, values for the total study population will be extracted and assumed to be similar for randomised groups. For follow-up blood pressure values, we will aim to include mean levels during the follow-up period. If this is not reported, we will calculate mean levels for the full follow-up manually when several values are presented, or include values at a single time point (eg, last visit). Blood pressure differences between groups will be recorded as reported, or calculated manually from follow-up values when not reported separately.

## Outcome data

We will extract the number of participants and events for each treatment arm in each study, whenever possible. If the number of events is not presented, we will estimate the number of events from the measure reported, for example, calculating backwards from incidence rates, or manually counting number of events from Kaplan-Meier curves.[34] We will use the definitions described above for each outcome (see the Outcome measures section).

## Missing outcome data

The number of participants lost to follow-up will be recorded for cardiovascular events and mortality outcomes, for each treatment arm, separately. It is common in cardiovascular trials to follow participants for mortality through national or regional registers even if participants drop out from clinical follow-up during the intervention. Loss to follow-up will be reported descriptively, and the impact of attrition on primary outcomes will be estimated in sensitivity analyses using different assumptions for the outcome of the participants lost to follow-up allowing for some uncertainty in these assumptions.[35 36]

## Risk of bias assessment

All eligible trials will be assessed for risk of bias using the Cochrane Collaboration's risk of bias assessment tool, second version (RoB2).[37] The RoB2 tool includes five domains relating to the randomisation process, deviations from the intended intervention, missing outcome data, measurement of the outcome and selection of reported results. Risk of bias will be assessed in relation to assignment of intervention (intention-to-treat).

The first two domains will be assessed on study level, as the same judgement apply for all outcomes. Trials judged to be at high risk of bias in the randomisation process will be excluded from further analyses. In the second domain, PROBE trials will be considered to be at low risk of bias as long as no specific concerns are noted. The third domain will be assessed for mortality outcomes and non-mortality outcomes separately, as clinical events are usually not available for participants lost to follow-up, whereas vital status may still be available through population registers. If the number of participants with missing outcome data are of the same magnitude as the number of events, this will raise some concerns, whereas high risk of bias will generally require asymmetric missingness of such magnitude it could plausibly impact effect estimates in a clinically important way. For the fourth domain, a blinded endpoint committee will be regarded as the appropriate way of assessing outcomes. Outcome data reported as adverse events have generally been assessed in an open fashion to allow safety monitoring. This gives rise to some concerns, but will not be judged as high risk of bias because cardiovascular outcomes are fairly objective, and the value of blinding in such circumstances is probably limited.[38] For the fifth domain, assessing potential bias from selection of reported results, deviations will be noted for specific outcomes. Assessment will be made by two investigators independently and discrepancies resolved by discussion or third-party involvement.

The risk of bias assessment will be incorporated in the overall assessment of the quality of evidence, within the Confidence In the results from Network Meta-Analysis (CINeMA) framework, as described under the subheading 'Confidence in network evidence'.

## Data synthesis
### Pairwise meta-analysis

In the presence of two or more studies comparing the same pair of interventions, we will synthesise results using random effects pairwise meta-analysis. The clinical heterogeneity (ie, differences in study and participant characteristics) of the studies within the same treatment comparison will be evaluated prior to synthesis. We will estimate the between-study variance ($\tau^2$) using the restricted maximum likelihood estimator. Inference on statistical heterogeneity will be based on $\tau^2$ as well as on the $I^2$ statistic. Relative risks will be calculated for all outcomes in all trials according the intention-to-treat principle, or the modified intention-to-treat principle, using complete cases, where follow-up data are not available for all randomised participants. We will also perform sensitivity analyses with different assumptions about participants with incomplete data (see the missing outcome data section).

### Evaluation of the transitivity assumption

Transitivity is the fundamental assumption of NMA allowing for valid indirect inference. If transitivity is violated, results from NMA would be invalid. To evaluate transitivity, we will compare the distribution of the potential effect modifiers across the available direct comparisons. Potential effect modifiers include baseline systolic blood pressure, as this has previously been associated with the effect of blood pressure lowering in general[3]; stable coronary artery disease, as this may be associated with greater benefit at low blood pressure levels[3]; diabetes

mellitus, which may be associated with adverse treatment effect at low blood pressure levels[29]; and chronic kidney disease, for which the cut-off where RAAS inhibitors are considered superior to other agents differ somewhat between different guidelines.[7 8] If the distributions of these characteristics are similar, we will infer against evidence of intransitivity. We will also assess the similarity of the node definitions (eg, comparability of drug doses) in studies making different treatment comparisons.

### Network meta-analysis

In the absence of important intransitivity, we will then perform random effects NMA. Two sets of networks will be created. First, drug classes will be grouped according to the conventional classification described above, each drug class representing one node in the network. Second, the diuretics node will be split into one thiazide-type node and one thiazide-like node. All outcomes will be analysed in both networks. As sensitivity analyses, we will analyse the primary efficacy and acceptability outcomes in both networks, combining the ACEi node and the ARB node into one RAAS-inhibitor node, as well as splitting the CCB node into dihydropyridine and non-dihydropyridine nodes, respectively. For each network, we will assume a common heterogeneity parameter across all comparisons.

Treatment effects will be presented for all active treatments compared with placebo, using forest plots, as well as for all comparisons within the decision set using league tables. Drug classes will be ranked, using the surface under the cumulative ranking curve (SUCRA), for each outcome separately in the two main networks, and for the primary efficacy and acceptability outcomes in the sensitivity analyses with combined RAAS inhibitors and subdivided CCB. SUCRAs for the primary efficacy and acceptability outcomes will be presented in two-dimensional graphs.

In case several studies evaluating drug class combinations are identified, we will perform a component-level analysis assuming each drug class forming an intervention component.[39 40]

Analyses will be performed in R using the netmeta package (Ref: https://cran.r-project.org/web/packages/netmeta/netmeta.pdf).

### Assessment of statistical incoherence

Incoherence will be assessed using a local and a global method: the Separating Indirect from Direct Evidence approach and the design-by-treatment interaction model, respectively. Given that tests for incoherence often have low power, we will consider p values smaller than 0.10 implying the presence of potential statistical incoherence.

### Subgroup analyses and meta-regression

Even if important intransitivity is not obvious for the potential effect modifiers listed above (see the Evaluation of the transitivity assumption section), potential differences in treatment effect based on the presence or absence of diabetes, coronary artery disease and kidney disease, will be explored in subgroup or meta-regression analyses. Although meta-regression is more powerful in detecting associations between trial characterises and treatment effect compared with subgroup analyses, our experience is that many trials do not report the exact percentage of participants with certain diseases. It may still be clear from the methods, or from other numbers presented, that it is well below certain thresholds. Thus, meta-regression will be used if permitted by the available data, otherwise trials will be dichotomised based on if ≥50% of the included participants had each characteristic at baseline. The impact of systolic blood pressure at baseline, as well as mean age at baseline, will be assessed through meta-regression analysis.

For some trials comparing different active treatments, the blood pressure level during follow-up will differ between treatment arms. From an intention-to-treat perspective, this is not a problem, since any class that reduces blood pressure more effectively would be expected to reduce the number of events to a larger extent. However, to be able to answer the question if any drug class has blood pressure-independent effects on any outcome, such differences need to be taken into account. This will be explored in meta-regression analysis, adjusting for mean blood pressure difference between comparisons within trials.

### Across-study bias

Comparison-adjusted funnel plots will be used to assess small-study effects for each outcome separately. If asymmetry is detected, we will employ network meta-regression models to test whether asymmetry is statistically significant and whether small studies tend to systematically favour specific drug classes.[41] For comparisons with 10 or more studies we will also use contour-enhanced funnel plots to investigate the possibility that asymmetry is due to publication bias.

### Confidence in network evidence

The overall confidence in the results will be assessed using the CINeMA framework.[42] This framework includes assessment of six domains; within-study bias, reporting bias, indirectness, imprecision, heterogeneity and incoherence. Taking the contribution of each trial for each comparison into account, the CINeMA tool provides a semiautomatic way of assessing the confidence in the results on comparison-level as well as on network-level.

### Ethics and dissemination

This review does not require ethical approval. The findings will be published in a peer-reviewed medical journal and presented at medical conferences.

**Contributors** MB conceived the idea. All authors (HJ, AC, BC and MB) contributed to the design of the study. HJ and MB drafted the manuscript. AC and BC revised the manuscript and contributed with important intellectual content. MB is the guarantor. All authors approved the final version.

**Funding** This work was supported by the Heart Foundation of Northern Sweden, with no specific grant number. The funder had no role in the design of the study.

**Competing interests**  None declared.

**Patient and public involvement**  Patients and/or the public were not involved in the design, or conduct, or reporting, or dissemination plans of this research.

**Patient consent for publication**  Not required.

**Provenance and peer review**  Not commissioned; externally peer reviewed.

**ORCID iD**

Mattias Brunström http://orcid.org/0000-0002-7054-0905

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
