## [Reviewer comments · BMJ Open]

ARTICLE DETAILS

TITLE (PROVISIONAL)	Comparative efficacy and acceptability of different antihypertensive drug classes for cardiovascular disease prevention – protocol for a systematic review and network meta-analysis
AUTHORS	Jussil, Heidi; Chaimani, Anna; Carlberg, Bo; Brunström, Mattias

VERSION 1 – REVIEW

REVIEWER	Christiana Kartsonaki University of Oxford
REVIEW RETURNED	18-Nov-2020

GENERAL COMMENTS	Comparative efficacy and acceptability of different antihypertensive drug classes for cardiovascular disease prevention – protocol for a systematic review and network meta-analysis This is a detailed protocol of a systematic review and network meta-analysis on antihypertensive drug classes for prevention of cardiovascular disease. The statistical methods are appropriate. Some minor comments: 1. There are a few typographical errors.2. P. 20, lines 1-2: “For this purpose, trials will be dichotomized based on if $\geq 50\%$ of the included participants had each characteristic at baseline.” Please explain the rationale for dichotomising in this way instead of using meta-regression with the percentage of participants having each characteristics as explanatory variable.3. How will trials with a high risk of bias be handled in the meta-analysis?
---

REVIEWER	Chau Ho Curtin University, Australia
REVIEW RETURNED	01-Dec-2020

GENERAL COMMENTS	Congrat the authors for a very detailed and well-prepared protocol for a systematic review and network meta-analysis. I have some minor comments as below: 1. Does the review include studies with different BP treatment strategy (e.g. intensive vs usual care)? How to manage the differences in BP treatment strategies among included trials?2. line 16 page 13: the label 'search strategy and study selection', could the authors change it to 'search strategy' as the study selection was described above in the section 'eligibility criteria'.
---

VERSION 1 – AUTHOR RESPONSE

Reviewer 1

- 1) Typos have been corrected throughout.
- 2) Thank you for this thoughtful comment. We agree that using the exact values for patient characteristics in meta-regression models would be more efficient compared to dichotomization. However, our previous experience from trials of antihypertensive treatment is that many trials do not report the exact number/percentage of participants with certain diseases. It may still be clear from the methods, or from other numbers presented, that it is well below 50%. For example, 30% previous CVD may be reported, which is not possible to convert into percentage with CAD in a reliable way, although it most certainly is below 50%. We have revised this section to the following:

Potential differences in treatment effect based on the presence or absence of diabetes, coronary artery disease, and kidney disease, will be explored in subgroup or meta-regression analyses. Although meta-regression is more powerful in detecting associations between trial characteristics and treatment effect compared to subgroup analyses, our experience is that many trials do not report the exact percentage of participants with certain diseases. It may still be clear from the methods, or from other numbers presented, that it is well below certain thresholds. Thus, meta-regression will be used if permitted by the available data, otherwise trials will be dichotomized based on if $\geq 50\%$ of the included participants had each characteristic at baseline.

- 3) The risk of bias assessment will be incorporated in the overall assessment of the quality of evidence, within the CINEMA framework, as described under the subheading “Confidence in network evidence”. This statement has been added under the “Risk of bias assessment” subheading for clarity.

Reviewer 2

- 1) We will not include trials assessing different strategies (e.g. intensive versus usual care), because such trials are not informative on the effect of different agents. We are aware of no trials assessing strategies versus agents, and thus inclusion of trials assessing different strategies would create two separate networks, without connecting nodes. This has been clarified under the subheading “Types of studies”:

Trials assessing different treatment strategies, such as more intensive versus less intensive treatment, will also be excluded, because they convey no information on drug- or drug class-specific effects.

- 2) This has been revised.